# High Quality Embeddings for Horn Logic Reasoning

**Yifan Zhang**                                                    YIZ521@LEHIGH.EDU
*Lehigh University, Computer Science and Engineering, 113 Research Dr., Bethlehem, PA, 18015*

**Yasir White**                                                    Y.WHITE005@GMAIL.COM
*Los Angeles Pierce College, Computer Science, 6201 Winnetka Ave, Woodland Hills, CA 91367*

**Dean Clark**                                                     DMC227@LEHIGH.EDU
*Lehigh University, Computer Science and Engineering, 113 Research Dr., Bethlehem, PA, 18015*

**Joseph Sanchez**                                                 JRS225@LEHIGH.EDU
*Lehigh University, Computer Science and Engineering, 113 Research Dr., Bethlehem, PA, 18015*

**Jevon Lipsey**                                                   JEVONLIPSEY1029@GMAIL.COM
*Colorado College, Computer Science, 14 E Cache La Poudre St, Colorado Springs, CO 80903*

**Ashely Hirst**                                                   ASH320@LEHIGH.EDU
*Lehigh University, Computer Science and Engineering, 113 Research Dr., Bethlehem, PA, 18015*

**Jeff Heflin**                                                    HEFLIN@CSE.LEHIGH.EDU
*Lehigh University, Computer Science and Engineering, 113 Research Dr., Bethlehem, PA, 18015*

**Editors:** Leilani H. Gilpin, Eleonora Giunchiglia, Pascal Hitzler, and Emile van Krieken

## Abstract

Neural networks can be trained to rank the choices made by logical reasoners, resulting in more efficient searches for answers. A key step in this process is creating useful embeddings, i.e., numeric representations of logical statements. This paper introduces and evaluates several approaches to creating embeddings that result in better downstream results. We train embeddings using triplet loss, which requires examples consisting of an anchor, a positive example, and a negative example. We introduce three ideas: generating anchors that are more likely to have repeated terms, generating positive and negative examples in a way that ensures a good balance between easy, medium, and hard examples, and periodically emphasizing the hardest examples during training. We conduct several experiments to evaluate this approach, including a comparison of different embeddings across different knowledge bases, in an attempt to identify what characteristics make an embedding well-suited to a particular reasoning task.

## 1. Introduction

Neural networks can be trained to rank the choices made by logical reasoners, resulting in more efficient searches for answers. Several researchers have explored whether machine learning can be used to improve the performance of first-order reasoning, similar to how AlphaGo used deep learning to improve AI game playing (Silver et al., 2016). There have been some promising results, but there is still much to do. We hypothesize that the best way to make progress on the problem is to solve three subproblems: representation, learning strategy, and control strategy. To make the problem more amenable to study, we have restricted our initial investigation to the Horn fragment of first-order logic.

This work contributes to a general neurosymbolic approach for logical reasoning that has three key components: 1) An embedding model that maps logical statements to vectors, 2) A scoring model that represents the likelihood that a path leads to a successful answer, and 3) A guided reasoner, where the neural model scores the choices of a backward-chaining reasoner. Our prior work focused on improving the scoring model and guided reasoner; here, we attempt different training methods for the embedding model to learn more useful embeddings and increase the overall performance. This paper extends the work previously presented in a workshop (White et al., 2025).

In particular, our contributions are 1) Three improvements to the process for learning an embedding model for logical statements, 2) An evaluation of the embeddings' ability to improve the efficiency of reasoning using a downstream scoring model, and 3) An investigation into what makes a particular embedding useful across a variety of knowledge bases (KBs).

## 2. Background

Neurosymbolic AI seeks "to integrate neural network-based methods with symbolic knowledge-based approaches" (Sheth et al., 2023). This can include a broad range of topics, from generating embeddings of knowledge graphs to training a neural network to predict whether one logical statement entails another. One of the earliest attempts to combine logic rules and neural networks was KBANN (Towell and Shavlik, 1994). KBANN takes propositional Horn rules and directly encodes them into the neural network. Kijsirikul and Lerdlamnaochai (2016) train a neural network that can perform inductive learning on first-order logic data. However, their architecture only allows the input of data representing a conjunction of atoms, and the output is a set of classes. There is no way to incorporate axioms into their reasoning. Rocktäschel and Riedel (2017) trained a neural network to perform unification and apply a backward-chaining-like process to predict missing atoms in a KB.

There are reasoners that attempt to leverage LLMs. AlphaGeometry generates proofs to solve Olympiad geometric problems (Trinh et al., 2024). It uses forward inference to find conclusions from a starting premise, then uses a language model to generate auxiliary points and retries the forward inference if the current proof space fails. Rather than being query-driven, AlphaGeometry deduces new statements exhaustively. ReProver, another LLM-based theorem prover for LeanDojo, selects premises from large math libraries (Yang et al., 2023). ReProver scores the retrieved premises using an LLM. Inherent in using these approaches is having unlimited access to an LLM and the ability to fine-tune it to the needs of reasoning.

Other researchers have used various representations for logical statements, particularly in the context of automated theorem proving. Jakubův and Urban use an approach based on term walks of length three (Jakubův and Urban, 2017). They parse each logical statement and create a directed graph of it, then extract every sequence of three nodes from the graph and create a code for it. This code has one dimension for each possible sequence; thus for a vocabulary $\Sigma$, the vector must have $|\Sigma|^3$ dimensions. This approach does not scale to KBs that have many constants.

Crouse et al. (2021) propose a chain-based approach that starts with a graph similar to Jakubův and Urban, but extracts patterns from a clause that starts with predicates and ends with variables or constants. Each sequence is hashed using MD5, and then that value is further reduced to $d$ dimensions by the modulo operator. Negative clauses are represented by the concatenation of an additional $d$ dimensions. When creating patterns, all variables are replaced by the symbol "*". This models the semantics that a variable can match with any term, but not that if the same variable appears multiple times in an expression, it must match the same term in every occurrence. Furthermore, the hashing and modulo operator addresses the scalability problem of termwalk, but it does mean statements are placed in latent space at random, as opposed to based on some inherent notion of similarity.

Arnold and Heflin (2022) proposed that embeddings for logical atoms could be learned in a way that retains the semantics of variables. A core operation used by any first-order logic reasoning algorithm is unification, where $\text{Unify}(\alpha, \beta)$ returns a substitution $\sigma$ such that $\alpha\sigma = \beta\sigma$ or fails. A set of unifying atoms and another set of non-unifying atoms can be found procedurally. Using a neural network that optimizes triplet loss (Vassileios Balntas and Mikolajczyk, 2016), these atoms can be mapped into embeddings such that unifying atoms are near, and non-unifying atoms are far away. We describe Arnold and Heflin's approach and our improvements in subsequent sections.

## 3. Learning to Improve Logical Inference

Backward-chaining is an algorithm for Horn logic inference that operates by starting with a goal and systematically working backward through a series of rules and known facts to determine the conditions required to achieve the goal. Traditional backward chaining reasoners often rely on brute-force to explore potential solutions, which can lead to inefficiencies as the complexity and scale increases. Even relatively small knowledge bases of thousands of statements can result in searches of millions of nodes unless they have been carefully designed by a knowledge engineer. Our prior work looked at learning a scoring model to direct the search along promising paths and compared chainbased, termwalk, and unification as a means for solving this problem (Jia et al., 2023). This work improves on the previous unification approach to learn meaningful embeddings for logical statements.

Starting with a knowledge base of facts and rules, we use a forward-chaining reasoner to infer new facts from the existing information. From these new facts, we randomly substitute constants with variables and divide the resulting list into one hundred training queries and one hundred test queries. We then solve the training queries using a randomized backward chaining reasoner, exploring all possible paths to a solution within a predefined depth limit. For each node in the path, we assign $\langle goal, rule, score \rangle$ tuples to the results. Here, *goal* refers to the target query, *rule* denotes the logical statement(s) used in the proof, and *score* represents the solvability of a rule, with a score of 1 indicating a successful rule and 0 indicating a rule that does not lead to a solution.

First, we learn embeddings for atoms using triplet loss (Vassileios Balntas and Mikolajczyk, 2016). Triplet loss requires a set of $\langle anchor, positive, negative \rangle$ tuples and learns embeddings that place the *anchors* close to the *positive* examples, and far from *negative* examples. Arnold and Heflin's original approach for training the embedding model (Arnold and Heflin, 2022) involved generating a list of atoms at random, and for each atom, selecting

from the same list a unifying atom to serve as the positive example and an atom that does not unify to serve as the negative example. The *anchor*, *positive*, and *negative* atoms are mapped onto a 50-dimensional embedding space using a three-layer network that minimizes triplet loss.

Second, the $\langle goal, rule, score \rangle$ tuples are converted into vectors using the embedding model. Finally, supervised learning with a two-layer neural net is used to train a scoring model. Initial experiments have shown that the learned model can often significantly reduce backward-chaining search, sometimes by an order of magnitude or more. However, improvements have failed to materialize for some queries and even some KBs.

## 4. Our Approach

In this paper, we experiment with three techniques to improve the embedding model and its downstream performance. First, we increase the likelihood of generating more atoms with repeated terms. Second, we define a balanced triplet generation approach that ensures a good mix of *easy* and *hard* training examples. Third, we periodically train the model on samples with the highest loss (Harwood et al., 2017).

### 4.1. Generating Atoms with Repeated Terms

We define repeated term atoms (RTAs) as logical facts that are not produced frequently by uniform distribution, but have additional semantics that the embedding model should learn. For example, $loves(X, X)$ will unify with far fewer atoms than $loves(X, Y)$. Initially, with our uniform random generation, the likelihood that we had a second term repeat was $1/4(1/v+1/c)$, a computed probability of 2.6% when the number of variables ($v$) was 10 and the number of constants ($c$) was 200. We modify the anchor generation process to produce a repeated term with a fixed probability. If the random atom has an arity $\geq 2$ then the *repeat chance* determines the likelihood that the next term will be a repeat of a preceding one. For example, with a sample vocabulary, the prior embedding approach only generated 110 (out of $10,000$) atoms with a repeated constant or variable. By creating an additional repeat chance of 15%, we are able to produce $3.3\times$ as many atoms with a repeated term. Note, since this process only applies to atoms with arity $\geq 2$, the resulting repeated term atoms are less than 15% of all anchors. Experiments show that this 15% value improves the overall performance of our embedding model and guided reasoner.

### 4.2. Balanced Triplet Generation Based on Difficulty

We propose a novel method for generating triplets. First, we create positive (i.e., unifiable) and negative (i.e., non-unifiable) atoms of both easy and hard difficulty levels by modifying different components of an anchor. For example, given an anchor atom such as $mom(X, john)$, an easy positive atom can be created by making a small modification, such as replacing $X$ with a constant (e.g., $mary$), yielding $mom(mary, john)$. A hard positive atom, in contrast, is generated through a larger modification, such as replacing both $X$ and $john$ with new variables (e.g., $Y$ and $Z$), resulting in $mom(Y, Z)$. In general, to derive a positive atom from an anchor, we can substitute a variable with another variable or constant, or vice versa. The degree of structural change determines the difficulty: the

more extensive the modification, the harder the positive example becomes. On the other hand, the rules are inverted when generating negative atoms. A large modification, such as altering the predicate to one with a different arity, typically produces an easy negative, as the resulting atom is very different from the anchor and cannot be unified. In contrast, a small modification—such as replacing one constant with another—often yields a hard negative, since the modified atom may appear superficially similar to the anchor but is still non-unifiable. These subtle distinctions compel the model to learn fine-grained structural cues that distinguish between superficially similar but logically incompatible atoms. Tables 4 and 5 in the appendix provide an overview of the structural types of anchor atoms and the transformation policies used to generate examples. Specifically, Table 4 categorizes anchor atoms with unary and binary predicates into seven structural types, while Table 5 details the transformation policies used to create easy and hard positive/negative examples. These generated atoms form the basis for constructing training triplets.

Subsequently, we construct triplets of three difficulty levels by systematically combining each anchor with positive and negative examples. We define three categories: **Easy triplets** have the form ⟨anchor, easy positive, easy negative⟩: a positive example that is structurally and semantically close to the anchor, and a negative example that is clearly distinct. As a result, the anchor-positive distance is small, the anchor-negative distance is large, and the margin between them is wide. This strong contrast provides a reliable training signal, helping the model establish basic discriminative boundaries and laying the groundwork for learning more complex structural distinctions. **Medium triplets** are either of the form ⟨anchor, easy positive, hard negative⟩ or ⟨anchor, hard positive, easy negative⟩: one example is structurally close to the anchor, while the other is more distant. As a result, the margin between the positive and negative atoms is narrower than in easy triplets, requiring the model to make more nuanced distinctions. Finally, **Hard triplets** have the form ⟨anchor, hard positive, hard negative⟩, presenting a greater challenge for triplet loss. The hard positive is logically unifiable with the anchor but structurally distant, often involving extensive modifications. In contrast, the hard negative may be deceptively similar in surface form yet logically non-unifiable. This setup reverses the typical distance expectations: in some cases, the negative may even lie closer to the anchor than the positive. Such configurations compel the model to go beyond superficial patterns and internalize deeper logical and structural criteria. Our triplet generation balances the distribution of each of these three types: 40% easy, 50% medium, and 10% hard triplets. When a hard triplet cannot be generated for a given anchor (e.g., due to structural constraints), we return to producing a medium triplet to preserve the balance and diversity of training data.

The number of possible atoms depends on the vocabulary of the KB. We describe the vocabulary for $kb$ using $np_{kb}$, $nc_{kb}$, $nv_{kb}$ and $ma_{kb}$ to represent the number of predicates, number of constants, number of variables, and maximum arity. Then the maximum number of unique atoms is $np_{kb} * (nc_{kb} + nv_{kb})^{ma_{kb}}$. As this quantity grows, more data will be needed for the model to capture the semantics of the atoms. An important consideration is how many triplets should contain the same anchor. Our experiments have suggested that if this number is too low, the model lacks sufficient examples to learn the diverse, unifiable, and non-unifiable patterns associated with a given anchor. But if the number is too high, then there may be too few anchors to properly generalize to unseen anchors. We define the triplets related to a single, unique anchor as the "triplets per anchor" (TPA). We define a

target number of triplets to generate synthetically, and based on a desired number of TPA, the number of unique atoms changes. For instance, if our target for training is 500k triplets, and our desired TPA is 20, the embedding model will train on 25k unique anchors, each paired with 20 associated triplets.

### 4.3. Training on Hard Samples

The final improvement to the embedding model involves a technique that focuses on training with the hardest samples. In prior work, the training loss tended to flatten out early, so we focus training on semi-hard and hard samples, which improves the generalization of the model (Harwood et al., 2017). Similar improvements from training on hard samples have been observed in Convolutional Neural Networks through the work of Sahayam et al. (2023). After generating a synthetic dataset of a few hundred thousand triplets, we periodically use half of the set with the highest loss to represent our "hardest" samples. During training at every n-epochs, we validate the model's performance and continue refining it using a subset of the original generation of triplets, focusing on the training samples with the highest loss. This forces the model to learn and focus on difficult triplets that will help it solve queries quickly. As we train the embedding model on the hardest samples, the "hard" samples gradually become "easy" samples, and these newer easy samples will no longer appear in future subsets of hard samples collected every 10 epochs from the original synthetic dataset. Through this practice, our model is still successful in training on all synthetic triplets generated.

## 5. Evaluation

In this section, we evaluate the proposed changes to train the embedding model. First,we conduct a study to see how the changes impact the performance of the end-to-end system. Then we conduct an ablation study to determine the extent to which each change contributes to performance. Finally, we investigate how consistent the embedding process is and try to identify what makes an embedding useful for several KBs.

Table 1: Comparing reasoners on different KB sizes.

| Reasoner | Size | Mean Nodes | Median | Fails |
|---|---|---|---|---|
| Standard | 250 | 17,204.0 | 1998.7 | 0.6 |
| Previous Embeddings | 250 | 981.8 | 3.4 | 0.0 |
| New Embeddings | 250 | 76.2 | 3.2 | 0.0 |
| Standard | 375 | 33,459.5 | 2429.2 | 2.8 |
| Previous Embeddings | 375 | 129,393.9 | 11.0 | 2.0 |
| New Embeddings | 375 | 45,367.7 | 2.7 | 0.2 |
| Standard | 500 | 3,419,493.6 | 552,639.1 | 7.4 |
| Previous Embeddings | 500 | 8,481,922.1 | 2.8 | 7.0 |
| New Embeddings | 500 | 609,466.6 | 2.5 | 6.6 |

### 5.1. Reasoning Performance

To test how our proposed embedding approach impacts downstream tasks, we consider three different KB sizes: 250 statements, 375 statements, and 500 statements. For each size, we generate a set of 5 synthetic KBs, as performance can be very different between KBs of the same size. We use 200, 300, and 400 constants in the 250, 375, and 500 statement KBs, respectively. Since larger vocabularies require more training, we trained the 250KB using 100k triplets, and both 375KB and 500KB using 200k triplets.

We have three representative reasoners, 1) a standard backward-chaining reasoner (*Standard*), 2) a guided reasoner trained using the original unification embedding approach (*Previous Embeddings*), and 3) a guided reasoner trained using everything proposed in this paper (repeated terms, balanced triplet generation, hard samples) to improve embeddings (*New Embeddings*). Both the previous embeddings and new embeddings systems use the Min Goal control strategy of Schack et al. (2024).

We report our results in Table 1. For each of the 5 KBs, we generated 100 unique queries. We collected the mean nodes explored across the 500 queries (100 per KB) to obtain our *Mean Nodes* metric displayed in the table. To obtain the *Median* nodes metric, we averaged the median nodes explored across each of the 5 KBs. The rows for *Standard* and *Previous Embeddings* are results reported in Schack et al. Schack et al. (2024). The *New Embeddings* experiments were conducted under (almost) identical conditions. The only difference is in the maximum number of nodes before a query **fails**: 100,000,000 in Schack et al., while 1,000,000 for the *New Embeddings* reasoner. This adjustment allowed the experiments to be conducted in days as opposed to weeks. The reasoner only failed on queries at the 500 size, so this is the only result impacted. If we replaced the 1,000,000 values of all failed queries with 100,000,000 (assuming the worst case that they would still time out with the larger cap), we would get a mean of 4,389,466.6. This worst case is still twice as good as the previous embeddings and only 28% worse than Standard.

We make the following observations from the data. Across each size, the medians are smaller than the mean because of a few large outliers present in each knowledge base. The medians for the two embedding approaches are orders of magnitude smaller than those of the standard reasoner. Except for the 375 KBs, the previous embeddings and new embeddings have very similar medians. The results for the means show that the new embeddings are much better than those of the previous system, ranging between 7.18% and 32.5% of the mean nodes explored. This means there are fewer large outliers, possibly indicating that the system generalizes better. It also scales to larger KBs better, although it can still be impacted by the occasional outlier. For example, two of the size 500 KBs had significantly worse performance than the others.

### 5.2. Ablation Study

To understand how each of our proposed embedding modifications impacted performance, we conduct an ablation study. Each ablation was conducted with 5 different knowledge bases of size 250. We used 20 predicates with a maximum arity of 2, and 200 unique constants when generating each synthetic KB. To keep the environment of our experiments as constant as possible, we also utilize the same KB and testing queries across the baseline and each ablation. Our results are shown in Table 2. *Baseline* represents recent prior work on

Table 2: Ablation study results.

| Reasoner | Nodes Explored |
|---|---|
| Baseline | 981.8 |
| Hard Samples | 325.2 |
| Triplet Difficulty | 169.0 |
| Repeated Terms | 99.0 |
| All Improvements | 76.2 |

learned embeddings (Schack et al., 2024), *Triplet Difficulty* represents the balanced triplet generation optimizations, *Repeated Terms* represents the increase in repeated term atom generation, and *Hard Samples* represents our minimization of examples with the highest triplet loss.

We first focus on measuring improvements from our triplet difficulty generation approach. The result is a mean nodes explored of 169.0, showing an 82.8% decrease from prior work. This decrease represents a reduction in time and resources needed to answer a set of queries.

Next, we observe the improvements of an increase in repeated terms by no longer generating anchor atoms uniformly. The result is a mean nodes explored of 99.0, representing an 89.9% decrease from prior work. This is a parameter that could be fine-tuned to improve overall performance, but we have yet to identify a relationship for the ideal number of RTAs needed to achieve lower nodes explored, nonetheless, a small increase in RTAs tends to improve our results significantly.

Finally, we examine the effects of training our model on half of the original dataset with the highest triplet loss. We observed an average of 325.2 nodes explored, a notable 66.8% decrease. Although "hard examples" lags slightly in terms of improvements against the other ablations, it still shows promise. In future work, we will improve on this technique by training on the most valuable samples with triplet mining, as opposed to training on the hardest half of examples (Harwood et al., 2017).

### 5.3. Embedding Consistency

In this section, we analyze the degree to which random effects in the embedding process impact the quality of downstream reasoning. We investigate the extent to which an embedding model generalizes to different knowledge bases built from a common vocabulary. The prior section only considered a single embedding model with each knowledge base. Here, we start with a similar setup: from a synthetic vocabulary, 5 KBs were created, with two different KB sizes: 250 statements and 500 statements. 200 and 400 constants were used for the 250 and 500 statement KBs respectively, and trained on 100,000 and 200,000 triplets as in the prior experiments. Both had a maximum number of nodes per query set to 1,000,000. This can lead to significantly fewer mean nodes than when run with a larger maximum value.

Where this experiment diverges is in training the guided reasoners. For each vocabulary, five embedding models were learned, and for each KB, five scoring models were created by using each embedding model. This leads to a total of 25 guided reasoners trained using our full set of embedding improvements. Using 100 training and 100 test queries, each

embedding/KB combination was evaluated based on the minimum, maximum, mean, and median nodes explored calculated from the means of the test queries on each KB.

Table 3: Summary of embedding model performance averaged over 5 KBs. (a) When applied to KBs with 250 statements (b) when applied to KBs with 500 statements.

$(a)$

| Model | Min | Max | Mean | StdDev | TV Dist | Sem Match |
|---|---|---|---|---|---|---|
| Model A | 2.6 | 3503.2 | 708.7 | 1562.3 | 0.15967 | 60.23% |
| Model B | 8.0 | 11374.4 | 2387.0 | 5028.7 | **0.17452** | 50.56% |
| Model C | 2.6 | **548.1** | **154.7** | **225.7** | 0.17349 | **62.30%** |
| Model D | 2.6 | 3724.7 | 762.7 | 1656.0 | 0.13722 | 61.97% |
| Model E | 2.6 | 570.2 | 231.3 | 306.4 | 0.13094 | 54.75% |

$(b)$

| Model | Min | Max | Mean | StdDev | TV Dist | Sem Match |
|---|---|---|---|---|---|---|
| Model F | **260.7** | 119111.9 | **33344.1** | 49566.0 | 0.15832 | **36.39%** |
| Model G | 486.2 | 725942.3 | 175125.3 | 309852.3 | 0.16379 | 25.69% |
| Model H | 559.7 | 159248.7 | 42983.8 | 65950.9 | 0.13459 | 29.45% |
| Model I | 300.5 | 113957.2 | 35593.8 | 46750.3 | **0.16532** | 35.60% |
| Model J | 983.6 | **107451.0** | 34093.2 | **44295.1** | 0.14027 | 28.14% |

The average of all knowledge bases is shown to give a relative example of the difficulty of each knowledge base for the reasoners. There is a large variation in the individual results: a 897x difference between the easiest and hardest knowledge base for KB250, and a 284x difference between the easiest and hardest knowledge base for KB500 (results for individual KBs are not shown). As shown in Table 3, when comparing the average nodes per query across embedding models, with KB250, Embedding Model C performs the best on average, performing the best for KB2, KB3, and KB5, about middle of the road for KB1, and significantly worse on KB4 than any other embedding model (548.12 nodes per query on average compared to 2.6 nodes per query for the other embeddings on average). None of the queries failed for KB250, showing that the only outliers are queries that could not easily find a solution using the guided reasoner. However, for KB500, this is not the case, as only 9 embedding/KB combinations had no queries failed, with KB4 being the only individual knowledge base where all the embedding models could solve each test query. That being said, Embedding Model F is slightly better than the others, having the best performance for KB1, KB4, and KB5.

To try to understand more about why the embedding model performs better or worse on a given knowledge base, we analyzed a few different properties of the embeddings. In each table, we have included columns for TV Distance and Semantic Match.

The Total Variation (TV) distance is a statistical metric that quantifies the difference between two probability distributions. In the context of embedding models, it can be used to measure how dissimilar the similarity score distributions are for positive and negative atoms in a test set. The TV distance ranges from 0 (identical distributions) to 1 (completely disjoint distributions). Models B and I had the most distinct distributions.

Semantic Match measures the amount of "leakage" in the scoring model from the training set to the test set. We use Jaccard Similarity to quantify the overlap between the scoring

models' training rule/goal set and the subgoals selected when executing the test queries. We calculate the similarity when treating two atoms that differ only by a variable renaming as equivalent. This is because these atoms are functionally no different during reasoning, even though it is possible that our embedding models will give them different scores. We also computed the exact match (not shown), and there were half as many matches under the stricter criterion.

From Table 3, embedding models C and F are the best performing across KBs, and these models also have the highest semantic match. However, the second-best performing models E and J have considerably lower semantic match values than the best performers. For TV distances, the results are even more inconclusive. Model C has a very high value, but Model F has a median value. The second-best performers, E and J, have the lowest value and the second-lowest value, respectively. These trends lead us to believe that neither TV distance nor semantic match is the sole determiner of a truly generalized embedding model, and potentially factors such as representation structure and predicate diversity could be other places to explore further. Overall, while there were general patterns found for the results from the embedding model/KB cross-testing do not entirely explain how well an embedding model can generalize over multiple sets, leaving solid avenues to explore in further studies to find a true singular number to quantify an efficient embedding.

## 6. Conclusion and Future Work

Our work supports researchers in reducing the computational cost of solving logical queries and constructing mathematical proofs. By focusing on approaches to train an embedding model, we have shown that while working on methods for scoring and choosing queries is important, there are downstream benefits by improving the embeddings for logical inference. The choice of representation for symbolic atoms and the process for learning these representations can have a significant effect. To put it pithily, "representation matters." We argue that the right representation for logical atoms places unifying atoms close in latent space, and we have demonstrated three approaches to help us learn good representations: 1) intentionally oversample anchor atoms that have repeated terms; 2) generate difficulty-aware positive and negative atoms by modifying anchors to form easy, medium, and hard triplets; and 3) use a training process that regularly emphasizes the hard triplets.

The insights from this work also suggest several promising directions for future investigation. For example, we plan to explore three enhancements: 1) integrating difficulty-aware weighting into the loss function to prioritize informative examples; 2) using adaptive margin strategies that reflect structural similarity; and 3) adopting a curriculum learning approach that gradually introduces more challenging triplets as training progresses. Furthermore, we also aim to apply our approach to larger, real-world knowledge bases and extend it beyond Datalog-style logic to full first-order logic. By doing so, we hope to further validate the generality and robustness of our embedding design strategy in practical reasoning systems.

## 7. Acknowledgments

This work was partially conducted as part of an REU site supported by the National Science Foundation under Grant No. CNS-2051037.

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

## Appendix A. Example Generation Policies

Table 4 presents the mapping between anchor atom types and the transformation policies used to generate training examples of varying difficulty. Anchor atoms are categorized based on predicate arity (unary or binary) and the composition of their arguments (e.g., constants, variables, or duplicates). For each anchor type, applicable transformation policies (defined in Table 5) are listed for generating positive (+) and negative (–) examples at easy (E) and hard (H) levels. This mapping guides the construction of diverse and semantically meaningful triplets for training.

Table 5 provides a detailed overview of the policies used to generate positive (unifiable) and negative (non-unifiable) atoms from anchor atoms. Each policy defines a specific transformation, such as replacing a constant with a variable, altering the predicate, or swapping arguments, that results in either a positive or negative example, at varying levels of difficulty.

The table is divided into four main sections: 1) the policy name and its definition, 2) whether it is used to produce positive (+), negative (–), or mixed (±) examples, and 3) example outputs for both unary and binary predicates under easy and hard difficulty settings.

Easy examples are typically generated through small modifications (e.g., changing a single argument). In contrast, hard examples involve larger structural changes (e.g., modifying the predicate with a different arity or both arguments). For negative examples, this pattern

Table 4: Anchors and Their Applicable Policies for Generating Positive (+) and Negative (−) Examples at Easy (E) and Hard (H) Difficulty Levels

| Anchor | Definition | E+ | H+ | E− | H− |
|---|---|---|---|---|---|
| p6(c1, c2) | Binary, 2 consts | OCNV | TNV | $NPA_r$, $NPA_g$, $NAllA_g$ | OCNC, OADA, NP |
| p6(v1, c1) p6(c1, v1) | Binary, var + const | OCNV, OADA, OVNC, OVNV | TNV, TSNV | $NPA_r$, $NPA_g$, $NAllA_g$ | OCNC, NP |
| p6(v1, v2) | Binary, 2 vars | OADA, OVNC, OVNV | TNV, TNC, TNDV, TNDC, SWA | $NPA_r$, $NPA_g$ | NP |
| p6(v1, v1) | Binary, dup var | OVNC, OVNV | TNDV, TNDC, TNVC | $NPA_r$, $NPA_g$ | TNC |
| p6(c1, c1) | Binary, dup const | OCNV | TNDV, TNV | $NPA_r$, $NPA_g$, $NAllA_g$ | OCNC, NP |
| p2(c1) | Unary, one const | OCNV | ∅ | $NPA_r$, $NPA_g$ | OCNC, NP |
| p2(v1) | Unary, one var | OVNC, OVNV | ∅ | $NPA_r$, $NPA_g$ | NP |

is reversed: larger changes often lead to easy negatives, while small, deceptive modifications tend to produce hard negatives. This table serves as the foundation for constructing structured triplets for training the embedding model.

In our example generation process, we allow the reuse of variable names from the anchor atom, such as generating $p6(v1, c1)$ from an anchor like $p6(v1, v2)$. However, this differs from real-world reasoning systems, which apply standardization separately to rename variables before unification. This ensures that variables in different clauses or queries are renamed to be syntactically distinct, thereby avoiding accidental unification due to shared variable names. In future work, we plan to investigate further the impact of enabling or disabling standardization apart from the example generation process. Specifically, we aim to understand how this choice affects both the embedding learning task and downstream reasoning performance.

Table 5: Policy definitions and corresponding example transformations

| Policy | Definition | ± | Easy Examples | | Hard Examples | |
|---|---|---|---|---|---|---|
| | | | Unary | Binary | Unary | Binary |
| OCNV | one const→one var | + | p2(c1)→p2(v1) | p6(c1,c1)→p6(c1,v1) | ∅ | ∅ |
| OCNC | one const→new const | − | ∅ | ∅ | ∅ | p6(c1,c2)→p6(c1,c4) |
| OADA | one arg→dup of another | ± | ∅ | p6(c1,v1)→p6(v1,v1) | ∅ | p6(c1,c2)→p6(c1,c1) |
| OVNC | one var→new const | + | p2(v1)→p2(c1) | p6(v1,c1)→p6(v1,c1) | ∅ | ∅ |
| OVNV | one var→new var | + | p2(v1)→p2(v2) | p6(v1,c1)→p6(v2,c1) | ∅ | ∅ |
| NPA$_r$ | pred→new pred (≠ arity) | − | p2(c1)→p6(v2,c3) | p6(v1,v1)→p2(c4) | ∅ | ∅ |
| NPA$_g$ | pred, args→new (≡ arity) | − | p2(c1)→p1(v2) | p6(v1,v1)→p5(c4,v3) | ∅ | ∅ |
| NAllA$_g$ | all args→new const/var | − | ∅ | p6(v1,c1)→p6(v2,c3) | ∅ | ∅ |
| NP | pred→new pred (≡ arity) | − | ∅ | ∅ | p6(c1)→p2(c1) | p6(c1,c1)→p3(c1,c1) |
| TNV | 2 args→2 new vars | + | ∅ | ∅ | ∅ | p6(v1,c1)→p6(v3,v2) |
| TNC | 2 args→2 new consts | ± | ∅ | ∅ | ∅ | p6(v1,v1)→p6(c1,c2) |
| TNDV | 2 args→new dup var | + | ∅ | ∅ | ∅ | p6(v1,c1)→p6(v2,v2) |
| TNDC | 2 args→new dup const | + | ∅ | ∅ | ∅ | p6(v1,v2)→p6(c1,c1) |
| TNVC | 2 args→new var + const | + | ∅ | ∅ | ∅ | p6(v1,v1)→p6(v3,c1) |
| SWA | swap args | + | ∅ | ∅ | ∅ | p6(v1,v2)→p6(v2,v1) |

