# OpenReview forum: "High Quality Embeddings for Horn Logic Reasoning"
_nesyconf.org/NeSy/2025/Conference_Phase_2 — NeSy 2025 - Phase 2 Poster_

### Official Review · Reviewer_EBVy · 2025-07-08
**Embedding enhancements for efficient ML‐based Logical Reasoning**

**Rating:** 6
**Confidence:** 3

**Review:**

This paper presents three techniques to enhance the quality of embedding models to improve the performance of ML-based first-order logical reasoners.

The authors first compare their combined approach, using all three techniques, against a baseline embedding model and a guided reasoner trained using the original unification embedding approach on synthetic KBs of 250, 375, and 500 statements. They report substantial reductions in both the mean and median number of nodes explored while using their new embedding techniques. An ablation study then evaluates the effect of each technique’s individual contribution. To assess generalization, they repeat the evaluation across five different embedding models on two KB sizes (250 and 500 statements). They observe consistent gains but note that their metrics do not entirely explain how well an embedding model can generalize over multiple KBs. Their main contribution—experimenting with three different embedding techniques—only works well on the synthetic, toy-sized KBs they are trained on and does not provide significant evidence of overall generalized impact. However, it serves as evidence for future exploration of embedding techniques to improve ML-based logical reasoners.

Overall, this work makes a good methodological contribution to embedding-guided reasoning, but its practical impact remains to be validated on larger, real-world knowledge bases, and further experiments need to be done to provide strong evidence of the generalization of the effect of their embedding techniques on the reasoners’ performance.

**Anonymity:**

Remain anonymous

---

### Official Review · Reviewer_GXLg · 2025-07-08
**Nice study with a number of suggestions and some validation.**

**Rating:** 7
**Confidence:** 4

**Review:**

The authors investigate the construction of embeddings to help symbolic reasoning with Horn Logic; this is clearly a neural tool for symbolic methods and thus is clearly relevant. The paper describes a host of ideas, some of them small, some not so small, but overall the added value of all these proposals seems to produce a positive result during evaluation.

The text is easy to read, at times feeling a bit informal, but overall conveying the ideas adequately.

Minor problem: the first sentence of Section 6 is rather weird; perhaps better to write "Our results are significant as they reduce..." or something similar.

A small point: use final period in Expression (1).

**Anonymity:**

Remain anonymous